# The state of artificial intelligence in medical research: A survey of corresponding authors from top medical journals

Michele Salvagno[1], Alessandro De Cassai[2]*, Stefano Zorzi[1], Mario Zaccarelli[1], Marco Pasetto[1], Elda Diletta Sterchele[1], Dmytro Chumachenko[3,4], Alberto Giovanni Gerli[5], Razvan Azamfirei[6], Fabio Silvio Taccone[1]

1 Department of Intensive Care, Hôpital Universitaire de Bruxelles (HUB), Brussels, Belgium, 2 Sant'Antonio Anesthesia and Intensive Care Unit, University Hospital of Padua, Padua, Italy, 3 Department of Mathematical Modelling and Artificial Intelligence, National Aerospace University "Kharkiv Aviation Institute", Kharkiv, Ukraine, 4 Ubiquitous Health Technologies Lab, University of Waterloo, Waterloo, Canada, 5 Department of Clinical Sciences and Community Health, Università degli Studi di Milano, Milan, Italy, 6 Department of Anesthesiology and Critical Care Medicine, Johns Hopkins University School of Medicine, Baltimore, MD, United States of America

☯ These authors contributed equally to this work.
* alessandro.decassai@aopd.veneto.it, alessandro.decassai@gmail.com (AC)

**Data Availability Statement:** All relevant data are within the manuscript and its Supporting Information files.

## Abstract

Natural Language Processing (NLP) is a subset of artificial intelligence that enables machines to understand and respond to human language through Large Language Models (LLMs).. These models have diverse applications in fields such as medical research, scientific writing, and publishing, but concerns such as hallucination, ethical issues, bias, and cybersecurity need to be addressed. To understand the scientific community's understanding and perspective on the role of Artificial Intelligence (AI) in research and authorship, a survey was designed for corresponding authors in top medical journals. An online survey was conducted from July 13th, 2023, to September 1st, 2023, using the SurveyMonkey web instrument, and the population of interest were corresponding authors who published in 2022 in the 15 highest-impact medical journals, as ranked by the Journal Citation Report. The survey link has been sent to all the identified corresponding authors by mail. A total of 266 authors answered, and 236 entered the final analysis. Most of the researchers (40.6%) reported having moderate familiarity with artificial intelligence, while a minority (4.4%) had no associated knowledge. Furthermore, the vast majority (79.0%) believe that artificial intelligence will play a major role in the future of research. Of note, no correlation between academic metrics and artificial intelligence knowledge or confidence was found. The results indicate that although researchers have varying degrees of familiarity with artificial intelligence, its use in scientific research is still in its early phases. Despite lacking formal AI training, many scholars publishing in high-impact journals have started integrating such technologies into their projects, including rephrasing, translation, and proofreading tasks. Efforts should focus on providing training for their effective use, establishing guidelines by journal editors, and creating software applications that bundle multiple integrated tools into a single platform.

**Funding:** The author(s) received no specific funding for this work.

**Competing interests:** The authors have declared that no competing interests exist.

## Introduction

Artificial intelligence (AI) and machine learning systems are advanced computer systems designed to emulate human cognitive functions and perform a wide range of tasks independently. The giant leaps these systems provide are the possibility to learn and solve problems through autonomous decision-making if an adequate initial database is provided [1]. Natural Language Processing (NLP) represents a field within AI focused on enabling machines to understand, interpret, and respond to human language meaningfully.

One intriguing advancement within the realm of AI is the development of Large Language Models (LLMs), which are a subset of NLP technologies. They are characterized by billions of parameters, which allows them to process and generate human-like text, understanding and producing language across a wide range of topics and styles.Generative chatbots, like ChatGPT(Generative Pre-trained Transformer), Microsoft Copilot, or Google Gemini, enhance these models and offer an easy-to-use interface. These LLMs excel in natural language processing and text generation, making them invaluable for diverse applications. Specifically, they have been used in medical research for estimating adverse effects and predicting mortality in clinical settings [2–4], as well as in scientific writing and publishing [5]. Finally, domain-specific or fine-tuned modelsare models that undergo additional training on a specialized dataset and are tailored to specific areas of expertise. This allows these models to develop a deeper understanding of terminology, concepts, and contexts, making them more adept at handling tasks ina specific field.

Potential applications of AI, and more precisely LLMs, in scientific production, are vast and multi-faceted. These applications range from automated abstract generation to enhancing the fluency of English prose for non-native speakers and even streamlining the creation of exhaustive literature reviews [6, 7]. However, AI output is far from being perfect, as AI hallucination has been well described and documented in the current literature [8, 9]. Additional concerns include ethical, copyright, transparency, and legal issues, the risk of bias, plagiarism, lack of originality, limited knowledge, incorrect citations, cybersecurity issues, and the risk of info-demics [9].

In light ofAI's novel application in scientific production, it remains unclear to what extent the scientific community understands its inherent potentials, limitations, and potential applications. To address this, the authors designed a survey to examine the level of familiarity, understanding, and perspectives among contributing authors in premier medical journals regarding the role and impact of artificial intelligence in top scientific research and authorship. We hypothesize that, given the novelty of large language models (LLMs), researchers might not be familiar with their use and may not have implemented them in their daily practice.

## Methods

### Survey design

An online survey in this study was conducted using the SurveyMonkey web instrument (https://www.surveymonkey.com, SurveyMonkey Inc., San Mateo, California, USA). The survey protocol (P2023/262) was approved by the Hospitalo-FacultaireErasme–ULB ethical commission(Comitéd'Ethiquehospitalo-facultaireErasme–ULB, chairman: Prof. J.M. Boeynaems) on July 11th, 2023.

Two members of the survey team (M.S. and A.D.C.) performed a bibliographic search on April 19, 2023, on PubMed and Scopus, to retrieve any validated questionnaire on the topic using the following search string: [((Artificial Intelligence) OR (ChatGPT) OR (ChatBot))

AND ((scientific production) OR (scientific writing)) AND (survey)]. No existing surveys on the specific topic were found.

Therefore, the research team constructed the questionnaire under the BRUSO acronym to create a well-constructed survey [10]. The survey consisted of 20 single-choice, multiple-choice, and open-ended questions investigating individuals' perceptions of using Artificial Intelligence (AI) in scientific production and content. The full list of questions is available for consultation in English (S1 Appendix Content 1, Survey Questionnaire in English).

## Population of interest

The population of interest in this survey consisted of corresponding authors who published in 2022 in the 15 highest-impact medical journals (S2 Appendix Content 2), as ranked by the Journal Citation Report from Clarivate. In this survey, we used the Journal Impact Factor (JIF) as a benchmark to target leading publications in the research field. Originally developed by Eugene Garfield in the 1960s, the JIF is frequently employed as a proxy for a journal's relative importance within its discipline. It is calculated by dividing the number of citations in a given year to articles published in the preceding two years by the total number of articles published in those two years. The focus on the corresponding authors aimed to access a segment of the research community that is potentially at the forefront of research publishing and scientific production. For this survey, only the email addresses of the corresponding authors listed in the manuscript were sought and collected. Whenmultiple emails were listed as corresponding, only the first email for each article was collected.When no email addresses were found, no further steps were taken to retrieve them.No differentiation was made regarding the type of published article, except for excluding memorial articles dedicated to deceased colleagues. All other articles were included. The authenticity of the email addresses or their correspondence with the author's name was not verified. As a result, it was not possible to calculate the a priori sample size.

## Survey distribution plan

To enhance the survey's effectiveness, a pretest was performed in two phases. In the first phase, the survey team reviewed the entire survey, with particular attention to the flow and the order of the questions to avoid issues with "skip" or "branch" logic. The time required to complete the survey was estimated to be around four minutes. In the second phase,the survey was distributed for validation to a small subset of participants, which included researchers working at the Erasme Hospital, to identify any issues before distributing it to the general population of interest. Their answers were not included in the final data analysis.

UsingSurveyMonkey's email distribution feature, the survey link was disseminated to all collected email addresses of the corresponding authors. To minimize the ratio of non-responders, reminder emails were sent one, two, and three weeks after the initial contact, with a final reminder sent one month later. Responses were collected from July 13th, 2023, to September 1st, 2023. SurveyMonkey's web instrument automatically identifies respondents and non-respondents through personalized links, allowing for targeted reminders to only those who had not yet completed the survey. This system also automatically prevents duplicate responses.

## Statistical analysis

Descriptive statistics was used to provide an overview of the dataset. Depending on the nature of the variables the results are reported either as percentages or as medians with interquartile range (IQR). Comparison among percentages were performed with the chi-square test with a p-values significance threshold at 0.05. All statistical analyses were performed using Jamovi

(Jamovi, Sydney, NSW Australia, Version 2.3) and GraphPad Prism (GraphPad Software, Boston, Massachusetts USA,Version 10).

## Results

A total of 4,302 email addresses for inclusion in the survey were collected from the list of journals in the appendix. Survey data were collected from 13[th] July to 1[st] September 2023. Following the initial email outreach and four subsequent reminders, 222 emails bounced back, and 142 recipients actively opted out of participating.Of those who opened the survey link, 266 respondents answered the initial questions. However, some immediately declined to continue, resulting in 236(5.5% of the emails sent) participants who started the survey and were included in the final analysis upon response.

The geographical distribution and demographic data of 229 respondents are depicted in **Table 1**,.The United States and the United Kingdom were most prominently represented, accounting for 57 (24.9%) and 41 (17.9%) of respondents, respectively. In total, English-speaking nations (USA, UK, Canada, and Australia) accounted for 124 (54.1%) of respondents.

The role of 229 responders is represented in **Fig 1**. Physicians, research academics and research clinicians were equally represented, with 64 (27.9%), 65 (28.4%) and 67 (29.2%) responders, respectively. The other responders declared not to be classified as the aforementioned and explained themselves mainly as journalists, students, veterinarians, editors, and pharmacists.

Most of the respondents to this question reported moderate 93 (40.6%) or little 60 (26.2%) familiarity with AI tools. Only 13 (5.7%) indicated extensive familiarity.Following questions up to Q14 were answered by all participants except for the 10 individuals (4.4%) who indicated no prior knowledge of AI (resulting in their automatic exclusion from answering those specific questions). Notably, 9 (69.2%)out of 13 with extensive familiarity reported AI tool usage, compared to lower rates among 20 out of 93 (21.5%)with moderate and 5 out of 60 (8.3%)minimal familiarity ($p < 0.001$).

More than half of 229 respondents (130, 55%) published their first medical article over 15 years ago, while 31 (13.5%) did so within the last five years. The median Scopus H-index among respondents was 24 (IQR 13–42). No statistically significant correlations were identified between H-index, AI familiarity and AI usage ($p > 0.05$).

Only 2 participants ($< 1\%$), reported receiving specific training in AI for scientific production. Despite this, 55 (24.02%) out of 229 responders usedAI tools in scientific content creation.Of these, the majority (67.3%) used ChatGPT. Interestingly, among participants from the US($n = 57$), a notable difference exists between those who have used AI for scientific production($n = 8$, 14%) and those who have not ($n = 49$, 86%).Those who published the first medical article more than 15 years ago, also declared to have ever used AI tools for scientific production in a lesser amount than the ones who published the first medical article less than 15 years ago(23/130 [17.7%] vs. 32/99 [32.3%], $p = 0.01$).

As shown in **Fig 2**, besides ChatGPT, among the 55 responders who have already published using the aid of AI during the scientific production,Microsoft Bing and Google Bard were used by 8 (14.5%) and 2 (3.6%) of respondents, respectively. Other large language models comprised 5.0% of the usage. Various software tools, including image creation and meta-analysis assistant tools, were also reported to be used by 7 (12.7%) and 6 (10.9%), respectively. Other AI tools reported are mainly Grammarly, Image Analysis tools, and plagiarism-checking tools.

When the 55 respondents who already used AI tools were asked about the primary applications of AI, 55.6% reported using AI for rephrasing text, 33.3% for translation, and 37.78% for proofreading. The rate of AI usage for language translation was consistent across English and

**Table 1. Age,sex and country of survey respondents.**

| | | |
|---|---|---|
| Age (total responders: 229) | < 35 years | 27 (11.8%) |
| | 35–44 years | 63 (27.5%) |
| | 45–54 years | 58 (25.3%) |
| | years | 45 (19.7%) |
| | > 65 years | 36 (15.7%) |
| Sex (total responders: 229) | Male | 150 (65.5%) |
| | Female | 77 (33.6%) |
| | Chose not to disclose | 2 (0.9%) |
| Country (total responders: 229) | Argentina | 1 (0.4%) |
| | Australia | 10 (4.4%) |
| | Austria | 3 (3.3%) |
| | Belgium | 7 (3.1%) |
| | Benin | 1 (0.4%) |
| | Brazil | 1 (0.4%) |
| | Canada | 16 (7.0%) |
| | Chile | 2 (0.9%) |
| | China | 5 (2.2%) |
| | Croatia | 1 (0.4%) |
| | Denmark | 1 (0.4%) |
| | Egypt | 1 (0.4%) |
| | Finland | 1 (0.4%) |
| | France | 3 (1.3%) |
| | Gambia | 1 (0.4%) |
| | Germany | 2 (2.6%) |
| | India | 4 (1.7%) |
| | Iran | 2 (0.9%) |
| | Israel | 1 (0.4%) |
| | Italy | 19 (8.3%) |
| | Mexico | 1 (0.4%) |
| | Morocco | 1 (0.4%) |
| | Netherlands | 13 (5.7%) |
| | New Zealand | 3 (1.3%) |
| | Nigeria | 1 (0.4%) |
| | Norway | 2 (0.9%) |
| | Peru | 2 (0.9%) |
| | Philippines | 1 (0.4%) |
| | Poland | 1 (0.4%) |
| | Portugal | 2 (0.9%) |
| | Republic of Korea | 1 (0.4%) |
| | Slovakia | 1 (0.44%) |
| | South Africa | 2 (0.9%) |
| | Spain | 5 (2.2%) |
| | Sweden | 2 (0.9%) |
| | Switzerland | 5 (2.2%) |
| | Turkey | 1 (0.4%) |
| | United Kingdom | 41 (17.9%) |
| | United States of America | 57 (24.9%) |
| | Zimbabwe | 1 (0.4%) |

Notably,among those older than 65 years(n = 36), 25 (69.4%) of respondents hailed from English-speaking countries (USA, UK, Canada, and Australia), and 27 (75.0%) were male. However, this latter proportion decreased to 59.3% (16 responders) of the 27 individuals younger than 35 years.

non-English-speaking countries (94.4% vs 92.4%,p = 0.547). Additional applications such as draft writing, idea generation, and information synthesis were each noted by 24.4% of respondents.

In the survey, 8 of the 51 who answered this question (15.7%) admitted to using a chatbot for scientific work without acknowledgment.By contrast, 27 (11.9%)out of 226 are certain they

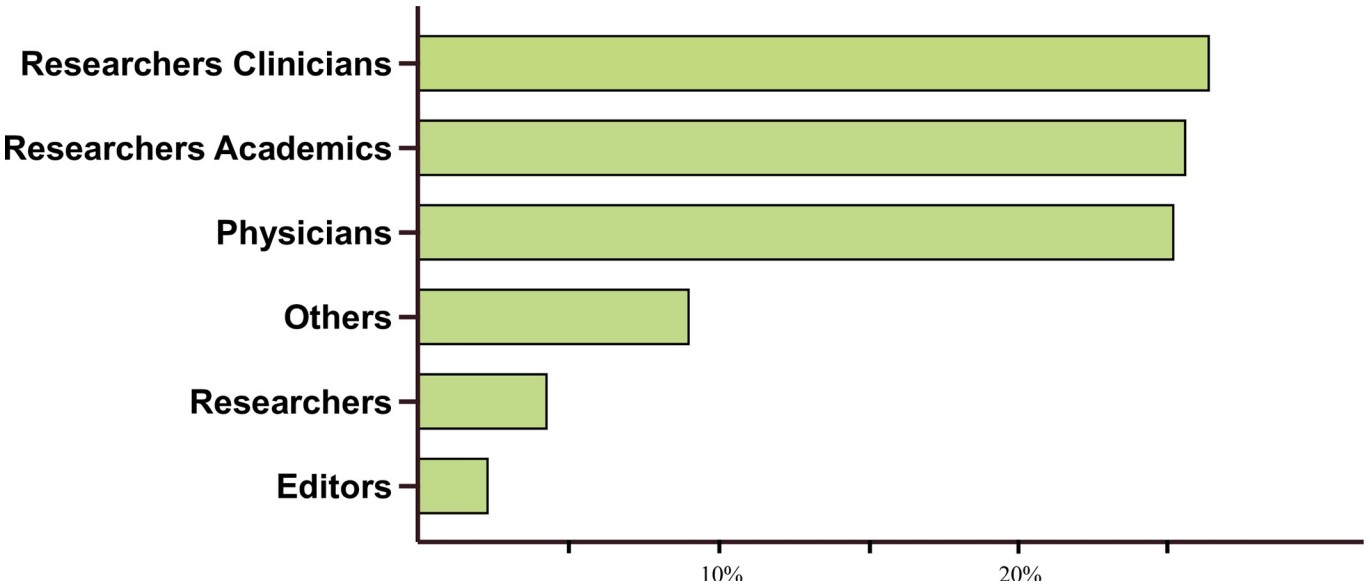

**Fig 1. Distribution of respondents by professional role.** Proportion of respondents in various professional roles as a percentage of the total respondent pool.

will employ some form of Artificial Intelligence in future scientific production. The complete set of responses is summarized in **Table 2**.

The primary challenges associated with utilizing AI in scientific research are outlined in Table 3.

The medical fields that respondents anticipate will gain the most from AI applications are Big Data Management and Automated Radiographic Report Generation. Additionalareas are detailed in Table 4.

When asked about their ability to distinguish between text written by a human and text generated by AI, 7 (3.1%) out of 226 respondents believed they could always tell the difference. Meanwhile, 120 (53.1%) felt they could only sometimes discern the difference. A total of 59

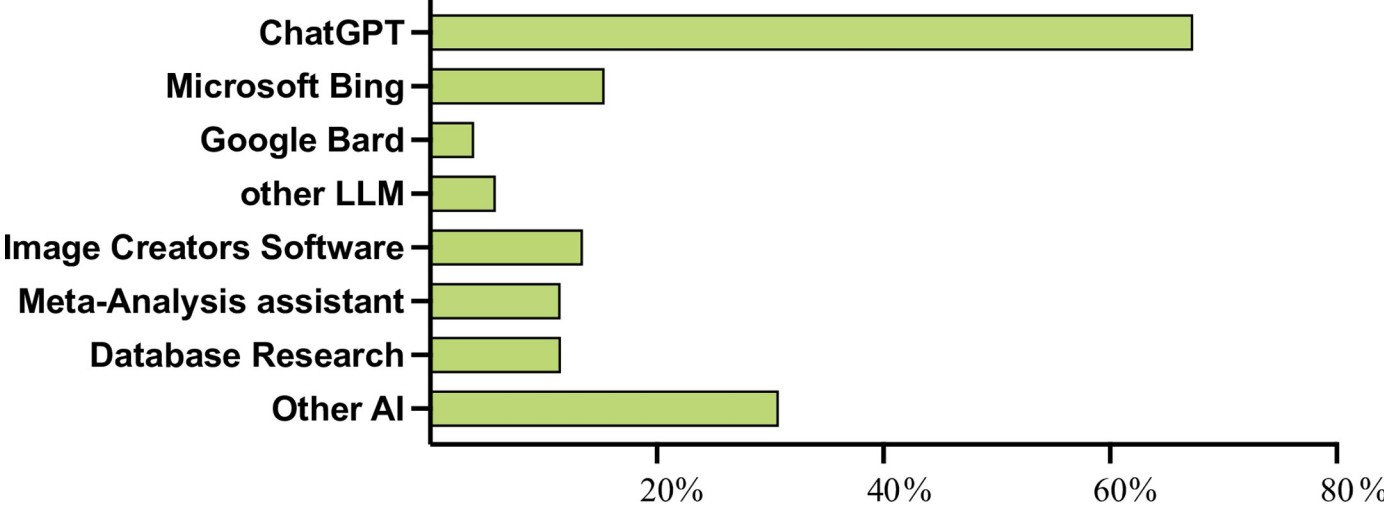

**Fig 2. AI tools used in scientific production.** The Y-axis lists the AI tools reported by respondents, while the X-axis shows their stated usage as a percentage. The total percentage exceeds 100% as respondents could report using multiple tools. LLM: Large Language Models; AI: Artificial Intelligence.

**Table 2. Future use of AI tools in scientific production.**

| Likelihood of Using AI in Future Scientific Production | Percentage (%) |
|---|---|
| Highly Improbable | 19 (8.4%) |
| Improbable | 21 (9.3%) |
| I Don't Know | 70 (31.0%) |
| Probable | 89 (39.4%) |
| Surely | 27 (11.9%) |

This table presents the intention of respondents to use AI tools in their future scientific work.

(26%)were uncertain, and a small fraction, 3 (1.3%), reported it is never possible to distinguish between the two.

Over 80% of respondents (n = 226) do not foresee AI supplanting the role of medical researchers in the future, with 81 (35.8%)strongly disagreeing and 106 (46.9%)disagreeing. A small fraction, 10 responders (4.4%), either somewhat or strongly agree that AI could take on the role of medical researchers. Meanwhile, 29 (12.8%)remain uncertain. By contrast, when it comes to the impact on clinical physicians,among the 226 responders to this last question, 177 (78.3%) anticipate that AI will partially alter the nature of their work within the next two decades. A minority of 18 responders (8.0%) foresee no change at all, and a very small fraction, 2 (0.9%), predict a complete transformation in the role of clinical physicians. To conclude, 14 (6.0%)are still unsure about the future impact of AI on clinical practice.

## Discussion

The present study aimed to explore the perceptions and utilization of Artificial Intelligence (AI) tools in scientific production among corresponding authors who published in the 15 most-impacted factor medical journals in 2022.

### Familiarity and training in AI

Intriguingly, this survey indicated that less than 1% of respondents had undergone formal training specifically designed for the application of AI in scientific research. This highlights a critical need for educational programs tailored to empower researchers with the necessary skills for effective AI utilization. The dearth of formal training may also contribute to the observed "limited" to "moderate" familiarity with AI concepts and tools among most survey participants, without a difference among ages and genders.Generally, AI tools are user-friendly and straightforward, requiring no specialized skills for basic usage. This could account for the

**Table 3. Key challenges in using AI for scientific production.**

| Factors | Responders, Percentage (%) |
|---|---|
| Technical Problems | 52 (23.0%) |
| Costs | 21 (9.3%) |
| Content Errors | 190 (84.1%) |
| Ethical Aspects | 150 (66.4%) |
| Privacy Aspects | 81 (35.8%) |
| Other | 30 (13.3%) |

This table outlines the primary challenges identified by respondents in utilizing AI tools for scientific endeavors. The total percentage exceeds 100%, as respondents (n = 226)could indicate multiple answers.

**Table 4. Medical fields expected to benefit most from AI applications.**

| Area of Medicine | Responders |
|---|---|
| Drug Development | 86 (38.1%) |
| Diagnosis | 108 (47.8%) |
| Treatment | 44 (19.5%) |
| Telemedicine | 63 (27.9%) |
| AutomatedRadiographic Report Generation | 126 (55.8%) |
| Big Data Management | 156 (69.0%) |
| Others | 24 (10.6%) |

This table highlights the medical specialties that respondents anticipate will gain the most from the integration of AI applications. The total percentage exceeds 100% as respondents (n = 226)could identify multiple fields.

lack of a significant difference between younger and older users.However, even though the basic use appears straightforward, a lack of comprehension may lead individuals to commit unnoticed errors with these tools, stemming from an unawareness of their own knowledge gaps [11].

Although beyond the primary focus of this study, we find it noteworthy to comment on the responses concerning the Scopus H-index. This score remains a subject of debate and is fraught with limitations, including self-citation biases, equal attribution regardless of author order and academic age,as well as gender-based disparities other than topic-specific biases. In our survey, the responders presented a median H-index of 24 (IQR 13–42), without statistically significant correlationsbetween H-index values and the variables of interest. Remarkably, two respondents indicated a lack of interest in monitoring their H-index. One respondent, a journal editor, expressed outright indifference with the remark "Who cares", probably echoing a sentiment that could be ascribed to Nobel Laureate Tu Youyou, whose current relatively low Scopus H-index of 16 belies her groundbreaking work on artemisinin, a treatment for malaria that has saved millions of lives.

## Applications of AI in scientific production

The survey results underscore a paradoxical relationship between familiarity with AI concepts and its actual utilization in scientific production. While many respondents indicated a "limited" to "moderate" familiarity with AI, around 25% reported employing AI tools in their research endeavors. This suggests that while the theoretical understanding of AI might be limited among the surveyed population, its practical applications are cautiously being explored. It is plausible that the rapid advancements in AI, coupled with its increasing accessibility, have allowed researchers to experiment with these tools without necessarily delving deep into the underlying algorithms and principles.Notably, the preponderance of the surveyed gravitated toward ChatGPT, suggesting a proclivity for natural language processing applications. Indeed, ChatGPT could assist scientists in scientific production in several ways [12].

The principal tasks for which AI was employed encompassed rephrasing, translation, and proofreading functions. AI tools, especially natural language processing models like ChatGPT, can significantly improve the fluency and coherence of scientific texts, especially for non-native English speakers. This is crucial in the globalized world of scientific research, where effective communication can determine the reach and impact of a study. Interestingly, the rates of AI use for language translation were quite similar between English-speaking and non-English-speaking countries, at 94.4% and 92.4%, respectively. This is unexpected since English is often the preferred language for communication in scientific fields, diminishing the

perceived need for translation tools. Several factors could explain this trend. First, these countries have a high proportion of expatriates, leading to many non-native English speakers in the workforce. One limitation of our study is that we did not inquire about the respondents' countries of origin, so we cannot provide further insights. Another possible explanation could be the selectivity of our respondent pool, which may not be sufficiently representative to show a difference in this variable.Nevertheless, ifthe predominant use of AI for tasks such as rephrasing, translation, and proofreading underscores its potential to enhance the quality of research output, it is essential to strike a balance to ensure that the essence and originality of the research are maintained in the pursuit of linguistic perfection.

This pattern intimates that, in its current stage, AI is predominantly perceived as a facilitator for enhancing the textual quality of scholarly work, rather than as an instrument for novel research ideation or data analysis. In response to this evolving landscape, academic journals, for example, JAMA and Nature, have issued guidelines concerning the judicious use of large language models (LLMs) and generative chatbots [13, 14]. Such guidelines often stipulate authors' need to disclose any AI-generated content explicitly, including the specification of the AI model or tool deployed.

While the survey highlighted the use of LLMs predominantly in textual enhancements, the potential of other AI in data analysis still needs to be explored among the respondents. Indeed, LLM and NLP, in general, currently have a very weak theoretical basis for data prediction.Nevertheless, longitudinal electronic health record (EHR) data have been effectively tokenized and modeled using transformer approaches, to integrate different patient measurements, as reported in the field of Intensive Care Medicine [15], even if this field is still insufficiently explored. Advanced AI algorithms can process vast datasets, identify patterns, and even accurately predict future trends, often beyond human capabilities. For instance, in biomedical research, numerous machine learning applications tailored to specific tasks or domains can assist in analyzing complex genomic data, predicting disease outbreaks, or modeling the effects of potential drugs. As indicated by the survey, the limited utilization of AI in these areas may be due to the lack of specialized training or apprehensions about the reliability of AI-generated insights.

## Future prospects

Most respondents were optimistic about the future role of AI in scientific production, with nearly 12% stating they would "surely" use AI in the future. This optimism towards integrating AI in scientific production can be attributed to the numerous advancements and breakthroughs in AI in recent years. As AI models become more sophisticated, their potential applications in research expand, ranging from data analysis and visualization to hypothesis generation and experimental design. The increasing availability of open-source AI tools and platforms makes it more accessible for researchers to incorporate AI into their work, even without extensive technical expertise.

However, most respondents (> 80%) did not believe that AI would replace medical researchers, suggesting a balanced view that AI will serve as a complementary tool rather than a replacement for human expertise. The sentiment that AI will augment rather than replace human expertise aligns with the broader perspective in the AI community, often termed "augmented intelligence" [16]. This perspective emphasizes the synergy between human intuition and AI's computational capabilities. While AI can handle vast amounts of data and rapidly perform complex calculations, human researchers bring domain expertise, critical thinking, and ethical considerations [17]. This combination can lead to more robust and comprehensive research outcomes [16, 18].

Moreover, the evolving landscape of AI in research also presents opportunities for interdisciplinary collaboration [19]. As AI becomes more integrated into scientific research, there will be a growing need for collaboration between AI specialists and domain experts. Such collaborations can ensure that AI tools are developed and applied in contextually relevant and scientifically rigorous ways. This interdisciplinary approach can lead to novel insights and innovative solutions to complex research challenges.

## Ethical and technical concerns

This survey identified a wide range of concerns regarding the integration of Artificial Intelligence (AI) into the realm of scientific research. Among these, content inaccuracies emerged as the most salient, flagged by over 80% of respondents. The risks associated with AI-generated content include creating ostensibly accurate but factually erroneous data, such as fabricated bibliographic references, a phenomenon described as "Artificial Intelligence Hallucinations"[20]. It has already been proposed that the Dunning-Kruger effect serves as a pertinent framework to consider the actual vs. the perceived competencies that exist regarding the application of AI in research [21]. Furthermore,the attitudes and expectations surrounding such technologies, just one year following the release of OpenAI's ChatGPT, can be aptly illustrated by the Gartner Hype Cycle [22]. Consequently, it is imperative that content generated by AI algorithms, even translations, undergo rigorous validation by subject matter experts.

Moreover, the rapid evolution of AI models, especially deep learning architectures, has created 'black box' systems where the decision-making process is not transparent [23]. This opacity can further exacerbate researchers' trust issues towards AI-generated content. The lack of interpretability can hinder the widespread adoption of AI in scientific research, as researchers might be hesitant to rely on tools they need to understand fully. Efforts are being made in the AI community to develop more interpretable and explainable AI models, but the balance between performance and transparency remains a challenge [24].

Beyond the ethical implications, another emerging concern is the potential for AI to perpetuate existing biases in the training data or continue "citogenesis"[25], which represents an insidious form of error propagation within the scientific corpus [26]. If AI models are trained on biased datasets, they can produce skewed or discriminatory results, leading to flawed conclusions and the perpetuation of systemic inequalities in research. This is particularly concerning in social sciences and medicine, where biased conclusions can have far-reaching implications [27]. For this reason, researchers must be aware of these pitfalls and advocate for the usage of data that is as unbiased and representative as possible in training AI models. The full spectrum of potential negative outcomes remains largely unquantified. Furthermore, using AI complicates the attribution of accountability, particularly in clinical settings. Ethical concerns, echoed by most of our respondents, coexist with legal considerations [28].

Additionally, integrating AI into scientific research raises data privacy and security questions [29]. As AI models often require vast amounts of data for continued training,there is the risk of submitted sensitive information being unintentionally exposed or misused during the process.This is one of the main reasons why several AI companies recently came out with enterprise and on-premise software versions.Such measures are especially pertinent in medical research, where patient data confidentiality is paramount [23, 30]. Ensuring robust data encryption and adhering to stringent data handling protocols becomes crucial when incorporating AI into the research workflow.

Various policy options have been tabled to govern the use of AI in the production and editing of scholarly texts. These range from a complete prohibition on using AI-generated content in academic manuscripts to mandates for clear disclosure of AI contributions within the text

and reference sections [31]. Notably, accrediting AI systems as authors appear to be universally rejected.Given these challenges, the concerns identified are legitimate and necessitate comprehensive investigation, particularly as AI technologies continue to advance and diversify in application.

A collaborative approach that includes AI experts, ethicists, policymakers, and researchers is crucial to manage the ethical and technical complexities and fully leverage AI in a responsible and effective manner. Furthermore, it is advisable for journal editors to establish clear guidelines for AI use, as some have already begun [14], including mandating the disclosure of AI involvement in the research process. Strict policies should be implemented to safeguard the data utilized by AI systems. Human oversight is necessary to interpret the data and results produced by AI. Additionally, an independent group should assess the impact of AI on research outcomes and ethical issues.

Lastly, attention must be paid to the energy consumption of AI systems and their consequent carbon footprint, which can be considerable, especially in the case of large-scale computational models [32]. AI and machine learning models, particularly those utilizing deep learning, require extensive computational resources and use significant amounts of electricity. To minimize this footprint, researchers should focus on optimizing AI algorithms to increase their energy efficiency and employ these systems only when absolutely necessary. It is essential for researchers to consider the environmental impact of their AI usage, treating ecological sustainability as a critical factor in today's world.

## Future in healthcare

The advent of AI in healthcare is rapidly evolving, and our responders anticipate Big Data Management [33] and Automated Radiographic Report Generation [34] to be the most impactful areas influenced by AI applications in the next few years. These results underline the growing recognition of AI's transformative potential in these domains [35]. Indeed, the current healthcare landscape generates massive amounts of data from diverse sources, including electronic health records, diagnostic tests, and patient monitoring systems [36]. AI-powered analytics tools could revolutionize how we understand and interpret this data, thus aiding in more accurate diagnosis and personalized treatment protocols. Similarly, medical imaging studies require considerable time and expertise for interpretation, representing a potential bottleneck in clinical workflow. Automated systems powered by AI can analyze images and rapidly generate reports with a speed and consistency that could vastly improve throughput and possibly contribute to improved patient outcomes, bolstering the assumption that AI-assisted radiologists work better and faster [37]. By contrast, these systems have been demonstrated to generate more incorrect positive results compared to radiology reports, especially when dealing with multiple or smaller-sized target findings [38]. Despite these and other limitations such as privacy security concerns, computer-aided diagnosis is promising and could impact several specialties [39]. In the market, there are already various user-friendly and easy-to-use mobile apps available, designed for healthcare professionals as well as patients, that offer quick access to artificial intelligence tools for obtaining potential diagnoses.Nevertheless, AI currently lacks the precision and capability to make clinical diagnoses, and thus cannot be a substitute for a doctor.

Finally, the development of AI in diagnosis and drug development was also highly rated in the survey. These results mirror current research trends, where AI has been applied for early disease detection and drug discovery processes, significantly cutting down time and costs. Even so, the essential human interaction between patient and clinician remains a core aspect of medical care, making it unlikely that AI will soon replace the need for in-person connection

[40]. Our survey respondents echo this sentiment, as the majority believe clinical doctors will only be partially replaced by technological advancements. Interestingly, in the open-ended responses, among the others, we found this comment "Humans do not want an AI-doctor". Even though literature tells us that AI could be more empathetic than human doctors [41], for the moment, everyone agrees.

## Limitations

While this study provides valuable insights into the understanding and utilization of Artificial Intelligence (AI) in scientific research, there are some noteworthy limitations. First, the study sample focuses exclusively on corresponding authors from high-impact medical journals. Although this allows us to capture perspectives from researchers at the forefront of scientific advancements, it may limit the generalizability of our findings to the broader scientific and medical community, including early-career researchers and students. Future surveys should aim to include a more diverse range of participants for a fuller picture.

Second, the survey had a low response rate. Physicians are generally challenging to be involved in survey research, and web-based surveys often yield lower participation rates [42]. Additionally, the accuracy of the email addresses is not guaranteed in email surveys, as evidenced by the emails that were bounced back, likely due to outdated or incorrect institutional email addresses. Nevertheless, although we didn't conduct an a priori sample size calculation, our aim was to collect responses from at least 300 participants to obtain a substantial perspective on the subject.

Third, the data was gathered through an online survey, which might introduce selection bias as those who are more comfortable with technology and AI may have been more inclined to participate.

Fourth, there was no verification process for the authenticity of the email addresses used in our study, which leaves room for potential inaccuracies in the data collected.

## Conclusions

This survey revealed varying degrees of familiarity with AI tools among researchers, with many in high-impact journals beginning to integrate AI into their work. The majority of respondents were from the USA and UK, with 54.1% from English-speaking countries. Only 5.7% indicated extensive familiarity with AI, and 24% used AI tools in scientific content creation, predominantly ChatGPT. Despite low training rates in AI (less than 1%), its use is gradually becoming more prevalent in scientific research and authorship.

## Supporting information

**S1 Appendix. Survey questionnaire.**
(DOCX)

**S2 Appendix. List of the leading 15 medical journals by impact factor.**
(DOCX)

## Author Contributions

**Conceptualization:** Michele Salvagno, Alessandro De Cassai.

**Data curation:** Michele Salvagno, Alessandro De Cassai.

**Formal analysis:** Alessandro De Cassai.

**Investigation:** Alessandro De Cassai.

**Methodology:** Michele Salvagno, Alessandro De Cassai.

**Supervision:** Fabio Silvio Taccone.

**Visualization:** Elda Diletta Sterchele.

**Writing – original draft:** Michele Salvagno, Alessandro De Cassai, Stefano Zorzi, Mario Zaccarelli, Marco Pasetto, Elda Diletta Sterchele, Dmytro Chumachenko, Alberto Giovanni Gerli, Razvan Azamfirei, Fabio Silvio Taccone.

**Writing – review & editing:** Michele Salvagno, Alessandro De Cassai, Stefano Zorzi, Mario Zaccarelli, Marco Pasetto, Elda Diletta Sterchele, Dmytro Chumachenko, Alberto Giovanni Gerli, Razvan Azamfirei, Fabio Silvio Taccone.

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
