## [Decision Letter · Decision Letter 0]

25 Mar 2024

PONE-D-23-38146The state of AI in medical research: a survey of corresponding authors from top medical journalsPLOS ONE

Dear Dr. De Cassai,

Thank you for submitting your manuscript to PLOS ONE. After careful consideration, we feel that it has merit but does not fully meet PLOS ONE’s publication criteria as it currently stands. Therefore, we invite you to submit a revised version of the manuscript that addresses the points raised during the review process.

We look forward to receiving your revised manuscript.

Kind regards,

Sanaa Kaddoura, Ph.D.

Academic Editor

PLOS ONE

Journal Requirements:

2. We note that Figure 1 in your submission contain [map/satellite] images which may be copyrighted. All PLOS content is published under the Creative Commons Attribution License (CC BY 4.0), which means that the manuscript, images, and Supporting Information files will be freely available online, and any third party is permitted to access, download, copy, distribute, and use these materials in any way, even commercially, with proper attribution. For these reasons, we cannot publish previously copyrighted maps or satellite images created using proprietary data, such as Google software (Google Maps, Street View, and Earth). For more information, see our copyright guidelines: http://journals.plos.org/plosone/s/licenses-and-copyright.

Reviewers' comments:

Reviewer's Responses to Questions

**Comments to the Author**

1. Is the manuscript technically sound, and do the data support the conclusions?

Reviewer #1: No

Reviewer #2: Partly

2. Has the statistical analysis been performed appropriately and rigorously? 

Reviewer #1: No

Reviewer #2: No

3. Have the authors made all data underlying the findings in their manuscript fully available?

Reviewer #1: Yes

Reviewer #2: Yes

4. Is the manuscript presented in an intelligible fashion and written in standard English?

Reviewer #1: Yes

Reviewer #2: No

5. Review Comments to the Author

Reviewer #1: This research entitled as “The state of AI in medical research: a survey of corresponding authors from top medical journals” has the following points to be addressed:

1. Abstract need to be concise and elaborate it clearly.

2. Title is not fully defended in the Abstract. Either change the title of the manuscript of explain it in the abstract.

3. Introduction part is also missing the context of the title i.e. AI in medical field is not fully defined.

4. The result phase showing the survey is related to scientific production and language translation etc. whether title showing the medical field.

5. Literature related to the field is not well organized.

6. Environmental concerns, problem statements and solution, future directions in the conclusion is missing.

Reviewer #2: The manuscript provides valuable insights into the state of AI in medical research, but it requires further improvement and enhancement. By addressing the below mentioned points, the authors can improve the clarity, rigor, and overall impact of the manuscript before publication.

1.Suggested Modification to the Title of the Manuscript and avoid short form in abstract e.g.AI

2. It is recommended to avoid the use of first-person pronouns (such as "I," "we," and "our") throughout the manuscript. Instead, use a more impersonal and objective tone by focusing on the subject matter rather than the author's perspective or involvement.

3.The abstract provides a concise summary of the manuscript, mentioning the purpose, methodology, and key findings of the survey. However, it could be enhanced by including more specific details about the results, such as the percentage of respondents familiar with AI and the specific tasks they integrated AI technologies into. Additionally, consider including a sentence highlighting the main implications or recommendations arising from the findings.

4.The introduction provides a clear rationale for conducting the survey and highlights the importance of understanding the role of AI in medical research. However, it could benefit from a more comprehensive review of the existing literature on AI in medical research.

5.Also, include key studies, advancements, or challenges in the field to provide a stronger context for the survey. Additionally, consider discussing the potential impact of AI in terms of improving patient outcomes, accelerating research processes, or addressing specific medical challenges.

6.The methodology section provides a general overview of the survey design and data collection process. However, it would greatly benefit from more detailed information. Include the specific survey questions like the integration of AI technologies, the challenges faced by researchers, etc.

7.Additionally, provide information on the response rate, demographics of the respondents, and any measures taken to ensure the representativeness of the sample. Including these details will increase the transparency and rigor of the study.

8.The results section presents the key findings of the survey, but it lacks in-depth analysis and interpretation. Consider providing more specific data, such as the percentage or frequency of responses for each survey question, to support the findings. Additionally, analyze the results in relation to the research questions or hypotheses posed at the beginning of the manuscript. Provide a more nuanced discussion of the variations in responses based on different factors, such as geographic location, years of experience, or journal impact factor. This will help readers better understand the implications and limitations of the findings.

9.The discussion section should expand upon the main findings and provide a comprehensive analysis. Start by reiterating the key findings and their significance. Then, delve into a more detailed discussion of the advantages, drawbacks, and challenges associated with integrating AI technologies in medical research, as mentioned in the abstract.

10.Include specific examples or case studies to illustrate the potential applications of AI and highlight the ethical considerations, bias mitigation strategies, and cyber security measures that need to be addressed.

11.Discuss how the findings of this survey align with or contribute to the existing literature on AI in medical research.

12.Finally, provide clear recommendations for future research, policy changes, or guidelines to support the effective and ethical use of AI in the field.

13.In conclusion, consider reiterating the key recommendations or action points mentioned in the discussion section. Emphasize the potential benefits of integrating AI technologies in medical research while acknowledging the challenges and the need for further research and development.

14.Avoid introducing new information or ideas in the conclusion.

15.Ensure that all references cited in the manuscript are included in the reference section and formatted correctly according to the journal's guidelines.

16.Additionally, consider expanding the list of references to include recent and relevant studies on AI in medical research.

6. PLOS authors have the option to publish the peer review history of their article (what does this mean?). If published, this will include your full peer review and any attached files.

Reviewer #1: No

Reviewer #2: No

---

## [Author Response · Author response to Decision Letter 0]

3 Jun 2024

Response to Reviewer #1

Q1. Abstract need to be concise and elaborate it clearly.

We have revised the abstract to be both concise and more explicit in its presentation.

Q2. Title is not fully defended in the Abstract. Either change the title of the manuscript or explain it in the abstract.

The Title now states: “The state of Artificial Intelligence in medical research: a survey of corresponding authors from top medical journals”. 

We also change the abstract.

Q3. Introduction part is also missing the context of the title i.e., AI in the medical field is not fully defined.

“AI in medical research” has been explained in the introduction.

Q4. The result phase showing the survey is related to scientific production and language translation etc. whether title showing the medical field.

The title has been adapted with “medical research”.

Q5. Literature related to the field is not well organized.

We have updated the literature review.

Q6. Environmental concerns, problem statements, and solution, future directions in the conclusion is missing.

Environmental concerns are a significant issue, and we have addressed this by adding a section on the topic. We have also incorporated the requested information as needed.

 

Response to Reviewer #2

1. Suggested Modification to the Title of the Manuscript and avoid short form in abstract e.g. AI.

We have revised the title to better reflect the focus of our study and expanded the term "AI" to "Artificial Intelligence" in the abstract to improve clarity.

2. It is recommended to avoid the use of first-person pronouns (such as "I," "we," and "our") throughout the manuscript.

That’s a fair point. We have edited the manuscript to eliminate first-person pronouns, adopting a more objective and impersonal tone throughout.

3. The abstract provides a concise summary of the manuscript, mentioning the purpose, methodology, and key findings of the survey. However, it could be enhanced by including more specific details about the results, such as the percentage of respondents familiar with AI and the specific tasks they integrated AI technologies into. Additionally, consider including a sentence highlighting the main implications or recommendations arising from the findings.

The abstract has been enhanced with specific details about the results, including the percentage of respondents familiar with AI. We have also added a sentence to summarize the main implications and recommendations of our findings.

4. The introduction provides a clear rationale for conducting the survey and highlights the importance of understanding the role of AI in medical research. However, it could benefit from a more comprehensive review of the existing literature on AI in medical research.

A bibliography has been updated with a more comprehensive review. 

5. Also, include key studies, advancements, or challenges in the field to provide a stronger context for the survey. Additionally, consider discussing the potential impact of AI in terms of improving patient outcomes, accelerating research processes, or addressing specific medical challenges.

The bibliography has been updated with a more recent review.

6. The methodology section provides a general overview of the survey design and data collection process. However, it would greatly benefit from more detailed information. Include the specific survey questions like the integration of AI technologies, the challenges faced by researchers, etc.

The specific survey questions have been added in the appendix as a supplement.

7. Additionally, provide information on the response rate, demographics of the respondents, and any measures taken to ensure the representativeness of the sample. Including these details will increase the transparency and rigor of the study.

More information has been provided to increase the study's transparency.

8. The results section presents the key findings of the survey, but it lacks in-depth analysis and interpretation. Consider providing more specific data, such as the percentage or frequency of responses for each survey question, to support the findings. Additionally, analyze the results in relation to the research questions or hypotheses posed at the beginning of the manuscript.

More information on these issues has been provided to increase the study's transparency.

9. Provide a more nuanced discussion of the variations in responses based on different factors, such as geographic location, years of experience, or journal impact factor. This will help readers better understand the implications and limitations of the findings.

The discussion verts on variations in responses based on different factors. 

10. The discussion section should include the main findings and provide a comprehensive analysis. Start by reiterating the key findings and their significance. Then, delve into a more detailed discussion of the advantages, drawbacks, and challenges associated with integrating AI technologies in medical research, as mentioned in the abstract.

Adetailed discussion of the advantages, drawbacks, and challenges associatedis provided.

11. Include specific examples or case studies to illustrate the potential applications of AI and highlight the ethical considerations, bias mitigation strategies, and cyber security measures that need to be addressed.

It is not in the scope of our paper. 

12. Discuss how this survey's findings align with or contribute to the existing literature on AI in medical research.

A section detailing the contribution of this survey to the existing body of literature on AI in medical research is included.

13. Finally, provide clear recommendations for future research, policy changes, or guidelines to support the effective and ethical use of AI in the field.

The section on ethical guidelines have been revised.

14. In conclusion, consider reiterating the key recommendations or action points mentioned in the discussion section. Emphasize the potential benefits of integrating AI technologies in medical research while acknowledging the challenges and the need for further research and development.

The conclusions have been rewritten.

15. Avoid introducing new information or ideas in the conclusion.

No new information has been introduced in the conclusion.

16. Ensure that all references cited in the manuscript are included in the reference section and formatted correctly according to the journal's guidelines. Additionally, consider expanding the list of references to include recent and relevant studies on AI in medical research.

This has been done, as requested.

We hope these comprehensive responses address the concerns raised and significantly enhance the manuscript's clarity and impact. Thank you for yourvaluable feedback

---

## [Decision Letter · Decision Letter 1]

30 Jul 2024

PONE-D-23-38146R1The state of Artificial Intelligence in medical research: a survey of corresponding authors from top medical journalsPLOS ONE

Dear Dr. De Cassai,

Thank you for submitting your manuscript to PLOS ONE. After careful consideration, we feel that it has merit but does not fully meet PLOS ONE’s publication criteria as it currently stands. Therefore, we invite you to submit a revised version of the manuscript that addresses the points raised during the review process.

We look forward to receiving your revised manuscript.

Kind regards,

Sanaa Kaddoura, Ph.D.

Academic Editor

PLOS ONE

Reviewers' comments:

Reviewer's Responses to Questions

**Comments to the Author**

1. If the authors have adequately addressed your comments raised in a previous round of review and you feel that this manuscript is now acceptable for publication, you may indicate that here to bypass the “Comments to the Author” section, enter your conflict of interest statement in the “Confidential to Editor” section, and submit your "Accept" recommendation.

Reviewer #2: All comments have been addressed

Reviewer #3: (No Response)

2. Is the manuscript technically sound, and do the data support the conclusions?

Reviewer #2: Yes

Reviewer #3: Yes

3. Has the statistical analysis been performed appropriately and rigorously? 

Reviewer #2: N/A

Reviewer #3: No

4. Have the authors made all data underlying the findings in their manuscript fully available?

Reviewer #2: Yes

Reviewer #3: Yes

5. Is the manuscript presented in an intelligible fashion and written in standard English?

Reviewer #2: Yes

Reviewer #3: Yes

6. Review Comments to the Author

Reviewer #2: (No Response)

Reviewer #3: I- Your abstract contains too much background information. A good abstract should briefly and clearly convey the impact of your research without unnecessary length. Here are some key points to improve your abstract:

i. Context and Specific Topic: Include a sentence that briefly describes the general topic and the specific topic of your

research. This helps set the stage for your study without delving into excessive background details.

ii. Central Question or Problem: Clearly state the central questions or problems your research addresses. Explain why

these questions are important. Are you examining a new topic, filling a gap in previous research, applying new

methods to existing data, or resolving a dispute in the literature?

iii. Research Methods: Summarize your research and/or analytical methods. This should be brief but informative enough to give readers an understanding of how you conducted your study.

iv. Main Findings: Highlight your main findings, results, or arguments. Focus on the most significant outcomes of your research.

v. Significance and Implications: Explain the significance or implications of your findings. Why do these results matter? How do they contribute to your field of study?

II- At the end of the introduction add a paragraph to describe the next sections. For example: The rest of the paper is organized as follows: Section 2 presents the method, Section 3 ....

III- Add a related work section after the introduction to show what others have done.

IV- What hypothesis you are trying to verify or what research questions you are answering. You have to clearly state this in the paper.

V- The survey start collection date is July 12 or 13? In the methods you have stated 13 while in the results 12.

VI- The results of statistical analysis described in the methodology section are not discussed in the results.

VII- Fix paper spacings for example in sentence two in the introduction "The giant leapthese ..." must be "The giant leap these...". This issue happens through out all the paper.

VIII- The conclusion is too short, you have to briefly discuss numbers from your findings. The conclusion of a research paper restates the research problem and summarizes your arguments or findings.

IX- The appendix you referred to in the paper was not attached with the submission.

7. PLOS authors have the option to publish the peer review history of their article (what does this mean?). If published, this will include your full peer review and any attached files.

Reviewer #2: **Yes: **Dr.Rahul A. Gujar

Reviewer #3: No

---

## [Author Response · Author response to Decision Letter 1]

2 Aug 2024

Reviewer #3: 

Q1: Your abstract contains too much background information. A good abstract should briefly and clearly convey the impact of your research without unnecessary length. Here are some key points to improve your abstract:

i. Context and Specific Topic: Include a sentence that briefly describes the general topic and the specific topic of your

research. This helps set the stage for your study without delving into excessive background details.

ii. Central Question or Problem: Clearly state the central questions or problems your research addresses. Explain why

these questions are important. Are you examining a new topic, filling a gap in previous research, applying new

methods to existing data, or resolving a dispute in the literature?

iii. Research Methods: Summarize your research and/or analytical methods. This should be brief but informative enough to give readers an understanding of how you conducted your study.

iv. Main Findings: Highlight your main findings, results, or arguments. Focus on the most significant outcomes of your research.

v. Significance and Implications: Explain the significance or implications of your findings. Why do these results matter? How do they contribute to your field of study?

A1: We thank the Reviewer for the time spent reviewing our article. We have attempted to shorten the introduction as per your suggestion. However, your first comment contradicts the feedback provided in the last round of revisions by Reviewer #2 (point 3). If further changes are required, we kindly request guidance from the Editor, as we are encountering conflicting comments.

Q2:II- At the end of the introduction add a paragraph to describe the next sections. For example: The rest of the paper is organized as follows: Section 2 presents the method, Section 3 ....

A2: We respectfully disagree with the Reviewers. It is standard practice in medical publishing to include methods, results, and discussion sections, making an additional descriptive paragraph unnecessary. Furthermore, research article introductions typically conclude with the study's main objective, as seen in our study, where it states:”To address this, the authors designed a survey to examine the level of familiarity, understanding, and perspectives among contributing authors in premier medical journals regarding the role and impact of artificial intelligence in top scientific research and authorship. We hypothesize that, given the novelty of large language models (LLMs), researchers might not be familiar with their use and may not have implemented them in their daily practice.”

Q3: Add a related work section after the introduction to show what others have done.

A3: It is the first work on this topic, for this reason there is not other literature on the topic. We tried to discuss at our best in the discussion section. If Reviewer believe that we missed some important information we would be delighted to further improve our discussion

Q4- What hypothesis you are trying to verify or what research questions you are answering. You have to clearly state this in the paper.

A4: We added our hypothesis at the end of the introduction that now reads:”To address this, the authors designed a survey to examine the level of familiarity, understanding, and perspectives among contributing authors in premier medical journals regarding the role and impact of artificial intelligence in top scientific research and authorship. We hypothesize that, given the novelty of large language models (LLMs), researchers might not be familiar with their use and may not have implemented them in their daily practice.”

Q5: The survey start collection date is July 12 or 13? In the methods you have stated 13 while in the results 12.

A5: July 13. Thank you for pointing out this typo.

Q6:The results of statistical analysis described in the methodology section are not discussed in the results.

A6: You are absolutely right and we removed the statistical analysis not used in ourresearch article

Q7: Fix paper spacings for example in sentence two in the introduction "The giant leapthese ..." must be "The giant leap these...". This issue happens through out all the paper.

A7: We would like to apologize with the Reviewer and we believe it could have been related to a compatibility problem among different software. We hope that now it is resolved

Q8: The conclusion is too short, you have to briefly discuss numbers from your findings. The conclusion of a research paper restates the research problem and summarizes your arguments or findings.

A8: We reformulated the conclusion as follows:

“CONCLUSIONS

This survey revealed varying degrees of familiarity with AI tools among researchers, with many in high-impact journals beginning to integrate AI into their work. The majority of respondents were from the USA and UK, with 54.1% from English-speaking countries. Only 5.7% indicated extensive familiarity with AI, and 24% used AI tools in scientific content creation, predominantly ChatGPT. Despite low training rates in AI (less than 1%), its use is gradually becoming more prevalent in scientific research and authorship.

“

Q9: The appendix you referred to in the paper was not attached with the submission.

A9:Provided

---

## [Decision Letter · Decision Letter 2]

8 Aug 2024

The state of Artificial Intelligence in medical research: a survey of corresponding authors from top medical journals

PONE-D-23-38146R2

Dear Dr. De Cassai,

We’re pleased to inform you that your manuscript has been judged scientifically suitable for publication and will be formally accepted for publication once it meets all outstanding technical requirements.

Kind regards,

Sanaa Kaddoura, Ph.D.

Academic Editor

PLOS ONE

Additional Editor Comments (optional):

Reviewers' comments:

Reviewer's Responses to Questions

**Comments to the Author**

1. If the authors have adequately addressed your comments raised in a previous round of review and you feel that this manuscript is now acceptable for publication, you may indicate that here to bypass the “Comments to the Author” section, enter your conflict of interest statement in the “Confidential to Editor” section, and submit your "Accept" recommendation.

Reviewer #3: All comments have been addressed

2. Is the manuscript technically sound, and do the data support the conclusions?

Reviewer #3: Yes

3. Has the statistical analysis been performed appropriately and rigorously? 

Reviewer #3: N/A

4. Have the authors made all data underlying the findings in their manuscript fully available?

Reviewer #3: Yes

5. Is the manuscript presented in an intelligible fashion and written in standard English?

Reviewer #3: (No Response)

6. Review Comments to the Author

Reviewer #3: (No Response)

7. PLOS authors have the option to publish the peer review history of their article (what does this mean?). If published, this will include your full peer review and any attached files.

Reviewer #3: No

---

## [Editor Report · Acceptance letter]

13 Aug 2024

PONE-D-23-38146R2 

PLOS ONE

Dear Dr. De Cassai, 

I'm pleased to inform you that your manuscript has been deemed suitable for publication in PLOS ONE. Congratulations! Your manuscript is now being handed over to our production team.

Kind regards, 

on behalf of

Dr. Sanaa Kaddoura 

Academic Editor

PLOS ONE